

# The ecological niche characteristics and interspecific associations of plant species in the alpine meadow of the Tibetan Plateau affected plant species diversity under nitrogen addition

Xuemei Xiang, Ke Jia De, Weishan Lin, Tingxu Feng, Fei Li and Xijie Wei

College of Animal Husbandry and Veterinary Science, Qinghai University, Qinghai Province, Xining, China

## ABSTRACT

**Background:** Plant species diversity is of great significance to maintain the structure and function of the grassland ecosystem. Analyzing community niche and interspecific associations is crucial for understanding changes in plant species diversity. However, there are few studies on the response of plant species diversity, species niche characteristics, and interspecific relationships to nitrogen addition in alpine meadows on the Qinghai-Tibet Plateau.

**Methods:** This study investigates the effects of different levels of nitrogen addition (0, 15, 30, 45, and 60 g N $m^{-2}$) on plant species diversity, functional group importance values, niche width, niche overlap, and interspecific associations in an alpine meadow.

**Results:** 1) Compared with the control (CK), the Shannon-Weiner index and species richness index significantly increased by 11.36% and 30.77%, respectively, with nitrogen addition at 30 g N $m^{-2}$, while both indices significantly decreased by 14.48% and 23.08%, respectively, at 60 g N $m^{-2}$. As nitrogen addition increased, the importance value of grasses showed an upward trend, whereas the importance value of sedges showed a decline. 2) The niche width of *Poa pratensis* L., *Elymus nutans* Griseb., and *Stipa purpurea* Griseb. are increased with higher nitrogen addition. As nitrogen addition increases, the niche overlap values also show a rising trend. At 60 g N $m^{-2}$, the overall community association in the alpine meadow exhibited a significant negative correlation. These findings suggest that grasses exhibit strong ecological adaptability under high nitrogen addition and gain a competitive advantage in spatial competition, increasing their niche width. Moreover, as nitrogen levels increase, the importance values of grasses rise significantly, and their ecological characteristics become more similar, resulting in reduced niche overlap among plant species. Furthermore, high nitrogen addition intensifies interspecific competition between grasses, sedges, and forbs, disrupting the original balance and reducing species diversity. These insights provide a valuable understanding of changes in species diversity and competitive dynamics in alpine meadow plant communities under high nitrogen addition.

Corresponding author
Ke Jia De, dekejia1002@163.com

# INTRODUCTION

Biodiversity is essential for maintaining functionality and delivering ecosystem services (*Pennekamp et al., 2018*). Plant diversity enhances an ecosystem's capacity to support various functions and processes, such as carbon sequestration, productivity, and nutrient accumulation (*Midolo et al., 2019*). Over the past century, agricultural and industrial activities have increased nitrogen compound emissions, contributing to elevated atmospheric nitrogen deposition in natural and semi-natural ecosystems worldwide (*Erisman et al., 2013*). Nitrogen deposition is one of the primary drivers of plant biodiversity loss in terrestrial ecosystems (*Midolo et al., 2019*). The response of plant community diversity to nitrogen deposition varies depending on factors such as the amount and duration of nitrogen input, nitrogen forms, climate factors, and vegetation types (*Humbert et al., 2016*; *Perring et al., 2018*; *Simkin et al., 2016*; *Smith, Schuster & Dukes, 2016*). Some studies have shown that nitrogen enrichment promotes plant growth and consequently alters community composition (*Liao et al., 2024*; *Liu et al., 2024*), while others have demonstrated that species richness generally decreases with higher nitrogen inputs (*Fraser et al., 2015*; *Midolo et al., 2019*). The mechanisms behind the decline in plant species richness under high nitrogen concentrations remain unclear (*DeMalach & Kadmon, 2017*; *Harpole et al., 2017*). Therefore, exploring the mechanisms driving the reduction of plant diversity under high nitrogen levels is crucial for understanding the complex effects of future climate change on plant communities and providing data and theoretical support for biodiversity conservation.

Some studies suggest that the decline in plant species richness under high nitrogen concentrations is due to increased interspecific competition (*DeMalach & Kadmon, 2017*). Nitrogen addition boosts plant productivity and canopy height, shifting competition among plant species from below-ground nutrient competition to above-ground competition for light. This shift means that taller or faster-growing species can capture more light per unit area compared to shorter species, intensifying competitive exclusion (*Niu et al., 2018*). In this context, interspecific competition emerges as one of the most important factors influencing changes in plant species diversity (*Mahaut et al., 2020*). Interspecific relationships have long been a cornerstone of community ecology and are essential for understanding species distribution, community assembly, and responses to environmental changes (*Ding & Ma, 2021*). Interspecific competition includes both the struggle for limited resources and the mechanisms that enable stable coexistence, which are often revealed through species' niche characteristics and interspecific associations (*Zwolicki et al., 2016*). Niche characteristics represent how species in a community utilize resources and their functional relationships with other populations. These characteristics can be categorized into niche width and niche overlap (*Litchman et al., 2007*). Niche width reflects a species' capacity to use resources and adapt to environmental conditions

(*He et al., 2022*), while niche overlap indicates the degree to which two species in the community share and compete for a given resource (*Gu et al., 2019*). Thus, studying niche characteristics is critical for understanding coexistence mechanisms and predicting community succession. Numerous experiments have shown that moderate nitrogen addition can alleviate resource limitations or reduce nutrient heterogeneity, which, in turn, decreases niche overlap and leads to fewer coexisting species (*Harpole et al., 2016*). The alleviation of resource limitations also increases above-ground biomass, shifting competition from nutrient acquisition to light capture, indirectly reducing niche overlap and breadth, thereby intensifying interspecific competition (*Yan et al., 2020*). Niche characteristics (niche width and overlap) reflect a species' ability to utilize resources and its role within the community, while interspecific association refers to the spatial co-occurrence of species, largely determined by their coexistence in a specific environment (*Su et al., 2015*). Interspecific association is a crucial characteristic of plant communities, influencing their formation and evolution (*Sfenthourakis, Tzanatos & Giokas, 2006*). Integrating niche characteristics with interspecies associations provides a more comprehensive and effective understanding of community composition, species interactions, and predictions of population dynamics in grassland ecosystems (*Hu et al., 2022*). Research has shown that as grassland ecosystems transition from low-stability pioneer communities to highly stable climax communities, positive correlations between species gradually increase (*Löffler & Pape, 2020*). Conversely, when grassland ecosystems are in an unstable state, negative correlations between species tend to rise (*Cassini, 2020*). Studies have indicated that in the later stages of artificial grassland vegetation recovery, species tend to occupy narrower niches, leading to stable communities and harmonious coexistence. Over time, the impact of interspecies associations on community stability increases, making it a valuable tool for quantifying community structure (*Wu et al., 2022*). Therefore, investigating the underlying mechanisms of species diversity changes in grassland ecosystems under nitrogen addition by examining both niche characteristics (niche overlap and width) and interspecies associations is of significant importance. As a globally significant ecological region, the alpine meadows of the Tibetan Plateau harbor a unique repository of alpine biodiversity and genetic resources, while also providing essential ecosystem services such as nutrient cycling regulation and carbon storage (*Dong et al., 2023*). Atmospheric nitrogen deposition has become an important part of global change due to human activities such as the burning of fossil fuels and the excessive use of fertilizers (*Erisman et al., 2013*). This article aims to explore the combined effects of nitrogen addition on plant community species diversity, niche characteristics, and interspecific associations, providing theoretical support for maintaining grassland species diversity under future climate change. Specifically, this study addresses the following questions: (1) How does nitrogen addition affect species composition and diversity in alpine meadow plant communities? (2) How do changes in plant species niche characteristics and interspecific associations influence species diversity in alpine meadows following nitrogen addition?

## MATERIALS AND METHODS

### Study area overview

The study was conducted at the Sanjiangyuan Ecosystem Field Observation Station of the Ministry of Education, located at Qinghai University (33°24′30″N, 97°18′00″E) at 4,270 meters. The region experiences a typical plateau continental climate. According to observations by the Chengduo County Meteorological Bureau in Yushu Prefecture from 2020 to 2022, the annual average temperature ranges from −10.3 °C to 4.6 °C, and the yearly precipitation averages 614.1 mm, with most rainfall occurring between June and September. The grassland is categorized as *Kobresia humilis* meadow, with dominant vegetation species including *Kobresia humilis* (C.A.Mey.ex Trauvt.) Sergiev. and *Kobresia pygmaea* Clarke of Cyperaceae, as well as *Elymus nutans* Griseb. and *Poa annua* L. of Gramineae. The soil in the study area is alpine meadow soil, with a pH of 6.92, organic matter content of 2.36%, available nitrogen content of 14.0 mg·kg$^{-1}$, available phosphorus content of 7.0 mg·kg$^{-1}$, and available potassium content of 76.5 mg·kg$^{-1}$.

### Experimental design

On June 25, 2020, a flat, representative area with uniform vegetation was selected as the experimental site in Zhenqin Town, Chengduo County. The experiment was arranged in a completely randomized block design. Based on previous research, the nitrogen addition threshold for moderately degraded grasslands on the Tibetan Plateau is 272 kg N ha$^{-1}$ year$^{-1}$ (*He et al., 2024*), and the nitrogen addition levels were set according to the conditions of the study area. Five nitrogen levels were established: 0 (CK), 15 g N·m$^{-2}$ (N15), 30 g N·m$^{-2}$ (N30), 45 g N·m$^{-2}$ (N45), and 60 g N·m$^{-2}$ (N60), corresponding to urea application rates of 32.60, 65.22, 97.83, and 130.43 g·m$^{-2}$. Each treatment had three replicates, with a plot size of 20 m$^2$ (4 m × 5 m) and a 1 m buffer between plots. Urea (Yuntianhua brand, total nitrogen ≥46.4%) was applied once in June 2020, and the experiment continued until 2022.

In mid-August 2022, during the peak growing season, vegetation surveys were conducted to collect data. Three replicate samples were taken using a quadrat method with a sample size of 0.5 m × 0.5 m. The survey recorded plant species richness (number of species), plant height, plant coverage, and above-ground biomass. Plant coverage was measured as the percentage of each species' projected area relative to the total plot area. Plant height was measured by selecting five plants per quadrat in their natural state. Above-ground biomass was collected by clipping all plants at ground level within the quadrat and drying them in an oven at 80 °C until a constant weight was achieved.

### Data analysis

#### Species diversity index

In the vegetation survey, a total of 17 plant species were recorded, belonging to nine families and 16 genera. Following the taxonomic and nomenclatural information provided by the Flora of China (http://www.iplant.cn/foc/), the plant species were classified into three functional types: grasses (Gramineae species), sedges (Cyperaceae species), and forbs

**Table 1 Importance values of plant species under different levels of nitrogen addition.**

| Species | Abbreviation | Plant functional types | Treatment (%) | | | | |
|---|---|---|---|---|---|---|---|
| | | | CK | N15 | N30 | N45 | N60 |
| *Stipa purpurea* Griseb. | Sp | Grasses | 6.25 | 5.74 | 8.90 | 8.99 | 6.82 |
| *Poa pratensis* L. | Pp | | 11.32 | 17.21 | 16.20 | 25.67 | 33.29 |
| *Elymus nutans* Griseb. | En | | 13.33 | 11.80 | 14.43 | 18.21 | 38.64 |
| *Kobresia humilis* (C.A.Mey.ex Trautv) | Kh | Sedges | 45.2 | 38.72 | 36.25 | 25.64 | 4.11 |
| *Gentiana futtereri* Diels et Gilg | Gf | Weeds | 3.9 | 2.91 | 2.70 | 2.66 | 3.20 |
| *Potentilla anserina* L. | Pa | | 2.62 | 3.20 | 2.35 | 2.50 | 2.71 |
| *Taraxacum lugubre* Dahlst. | Tl | | 2.46 | 3.27 | 2.51 | 3.80 | 3.93 |
| *Thalictrum petaloideum* L. | Tp | | 0.83 | 1.64 | 1.39 | 1.72 | 0.00 |
| *Lancea tibetica* Hook. f. & Thomson | Lt | | 1.79 | 2.05 | 1.47 | 1.95 | 1.60 |
| *Oxytropis ochrocephala* Bunge | Oo | | 3.17 | 8.69 | 2.77 | 2.54 | 2.45 |
| *Aster flaccidus* Bunge | Af | | 5.28 | 2.49 | 2.67 | 2.07 | 2.44 |
| *Anaphalis lactea* Maxim. | Al | | 1.30 | 0.00 | 1.57 | 1.54 | 1.12 |
| *Polygonum viviparum* L. | Pv | | 0.36 | 1.02 | 2.01 | 0.83 | 0.00 |
| *Pedicularis kansuensis* Maxim. | Pk | | 0.53 | 0.00 | 1.43 | 0.76 | 0.00 |
| *Gentiana straminea* Maxim. | Gs | | 0.00 | 0.42 | 1.05 | 0.89 | 0.00 |
| *Saussurea superba* Anthony | Ss | | 0.36 | 0.29 | 1.12 | 0.31 | 0.00 |
| *Ranunculus longicaulis* C.A.Mey.var.nephelogenes (Edgew.) L.Liou | Rl | | 1.05 | 0.31 | 1.04 | 0.37 | 0.00 |

(excluding Gramineae and Cyperaceae species) (*Han et al., 2022*; *Wen et al., 2020*) (Table 1). Specifically, the species were divided into three groups: three genus and three species of grasses, one genus and one species of sedges, and 13 species from 12 genera and seven families of forbs. The weed group included species from the Gentianaceae, Rosaceae, Asteraceae, Ranunculaceae, Scrophulariaceae, Leguminosae, and Polygonaceae families.

The importance value (IV) of plant species can reflect the degree of dominance of species in the community. It is calculated by the average value of the sum of relative height (RH), relative coverage (RC), and relative biomass (RB). The formula is as follows (*Zhang et al., 2022c*):

$$IV = \frac{RH + RC + RB}{3}$$

Note: Relative height = the average height of each species (functional group)/the sum of the average height of all species (functional group) in the quadrat;

Relative coverage = the sub-coverage of each species (functional group) in the quadrat/ the total coverage of species (functional group);

Relative biomass = the sum of the aboveground biomass of each species (functional group) in the quadrat/biomass of all species (functional group).

In this study, the commonly used species diversity indexes were species richness index (R), Shannon-Weiner index (H), and Pielou evenness index (J), and their calculation formulas were as follows (*Wen et al., 2020*):

$$R = S$$

$$H = -\sum_{i=1}^{s} Pi \ln Pi$$

$$J = H/\ln S$$

In the formula, S is the number of species and Pi is the relative importance value of species i.

### Niche characteristics

To analyze niche characteristics and overall interspecific associations, we selected species with an importance value of ≥1% based on their ranking. Only species that were present in all surveyed plots were considered for this analysis (*Du et al., 2024*; *Yuan & Wang, 2023*). Therefore, the study focused on ten key species: *Stipa purpurea* Griseb., *Poa pratensis* L., *Elymus nutans* Griseb., *Kobresia humilis* (C.A.Mey.ex Trautv), *Gentiana futtereri* Diels et Gilg, *Potentilla anserina* L., *Taraxacum lugubre* Dahlst., *Lancea tibetica* Hook. f. & Thomson, *Oxytropis ochrocephala* Bunge, *Aster flaccidus* Bunge (Table 1).

Niche width and niche overlap were calculated using Levins' niche width index (Bi) and Levins' niche overlap index (Oik), with the formulas as follows (*Genitsaris et al., 2020*):

$$B_i = \frac{1}{\sum_{j=1}^{r} \left(P_{ij}\right)^2}$$

$$O_{ik} = \frac{\sum_{j=1}^{r} P_{ij}P_{kj}}{\sum_{j=1}^{r} \left(P_{ij}\right)^2}.$$

In the formula, $P_{ij}$ and $P_{kj}$ represent the proportion of the important value of species i and species k in the j square to the sum of the important values of the species in all squares, respectively. Bi represents the utilization of resources and environmental adaptability of the species, and $O_{ik}$ represents the degree of overlap between the two species in resource utilization.

### Overall interspecific association

The overall correlation (VR) between plant species is greater than 1 indicates a positive correlation between species, less than 1 indicates a negative correlation between species, and VR = 1 indicates no correlation between species. The calculation formula is as follows (*Perelman, León & Oesterheld, 2001*):

$$VR = \frac{S_T^2}{Q_T^2} = \frac{\frac{1}{N}\sum\limits_{j=1}^{N}(T_j - t)^2}{\sum\limits_{i=1}^{s}(1 - P_i)}$$

In the formula, S is the total number of species in the survey plot, N is the number of quadrats, $P_i$ is the percentage of each species in the total population, $T_j$ is the total number of species in sample j, and t is the average value of species in all samples.

Statistical W = VR × N was used to test the significance of VR value deviating from 1. If there is no significant correlation between species, then W conforms to $\chi^2$. This distribution provides a 90% probability bound: $\chi^2_{0.95}$ (N) < W < $\chi^2_{0.05}$ (N).

### Interspecific relationship analysis

Spearman correlation test and association coefficient (AC) test were used to determine the interspecific association. The Spearman correlation coefficient is based on quantitative data and more sensitively reflects the interspecific relationship of plants. The formula is as follows (*Bishara & Hittner, 2012*):

$$r_s(ik) = 1 - \frac{6\sum\limits_{j=1}^{N}(x_{ij} - x_{kj})^2}{N^3 - N}.$$

In the formula, the Spearman correlation range is [−1,1]; positive value indicates positive correlation, negative value indicates negative correlation; n is the total plot, and $x_{ij}$ and $x_{kj}$ are the important values of species i and species k in plot j, respectively.

The correlation coefficient, namely AC, is calculated as follows (*Ma et al., 2022*):

If ab ≥ bc, then

$$AC = \frac{ad - bc}{(a + b)(b + d)}.$$

If bc > ad and d ≥ a, then

$$AC = \frac{ad - bc}{(a + b)(a + c)}.$$

If bc > ad and d < a, then

$$AC = \frac{ad - bc}{(b + d)(d + c)}.$$

The value range of AC is −1 ~ 1. The closer the AC is to −1, the stronger the negative connection is. The closer AC is to 1, the stronger the positive connection is. When AC = 0, the species is completely independent. When AC = 0.67, there is a strong positive coupling; 0.67 > AC > 0 indicates a weak positive association, AC = 0 indicates no association, −0.67 < AC < 0 indicates a strong negative association, and AC ≤ −0.67 also indicates a strong negative association.

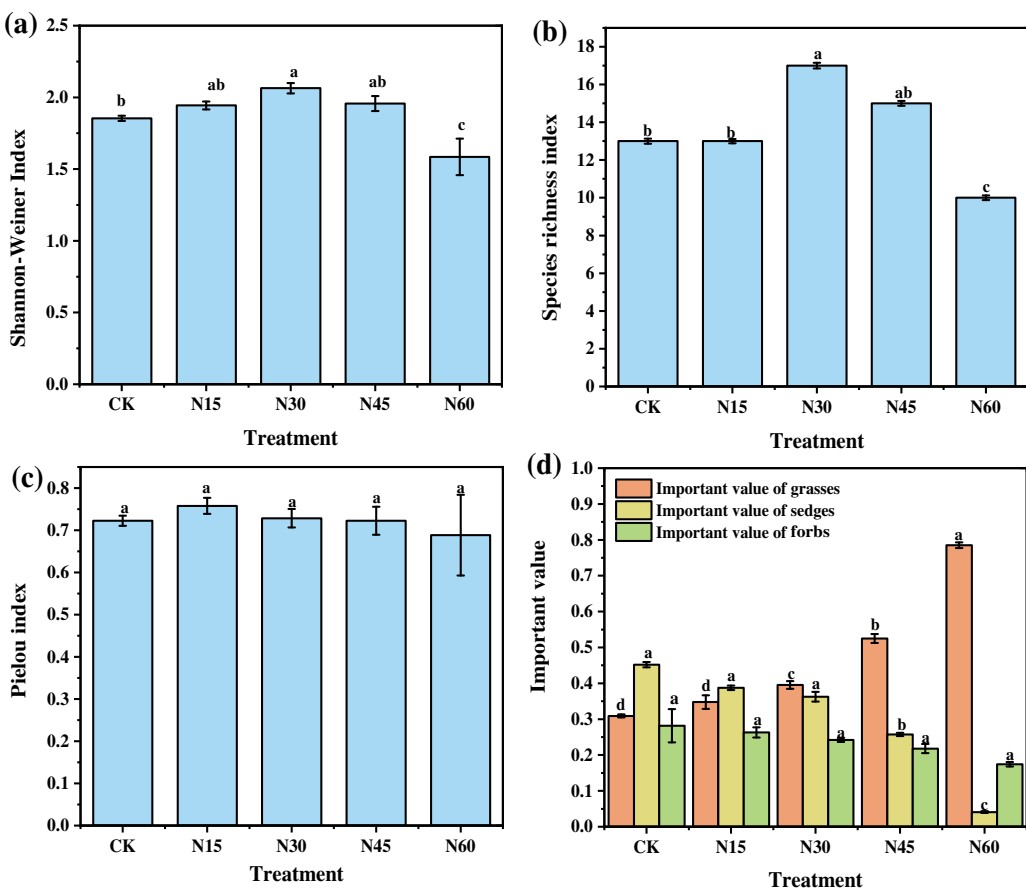

**Figure 1 Effects of nitrogen addition on species diversity index and functional group importance value of plant community.** (A) Shannon-Weiner index; (B) Species richness index; (C) Pielou index; (D) Important value of functional groups. Different letters indicate significant differences between N treatments at the 0.05 level. CK, N15, N30, N45, and N60 represent the treatments with no nitrogen addition, 15, 30, 45, and 60 g N m$^{-2}$, respectively. Data are presented as mean ± standard error.

## Statistical analyses

In the R 4.2.3 (*R Core Team, 2023*) version, the 'spaa' software package was used to calculate the species association index and niche. SPSS 25.0 software (SPSS Inc., Chicago, IL, USA) was used for one-way analysis of variance and least significant difference (LSD) *post hoc* test to test the effects of different levels of nitrogen addition on plant species diversity index, important value of functional groups and niche width, showing significance at a confidence level of 0.05. The significance test of the Spearman correlation coefficient was performed and plotted in Origin 2023b (Origin Lab Corp., Hampton, MA, USA).

## RESULTS

### Plant species diversity

The plant species diversity indices showed significant variations under different levels of nitrogen addition (Fig. 1). Compared to the control (CK), the Shannon-Weiner index

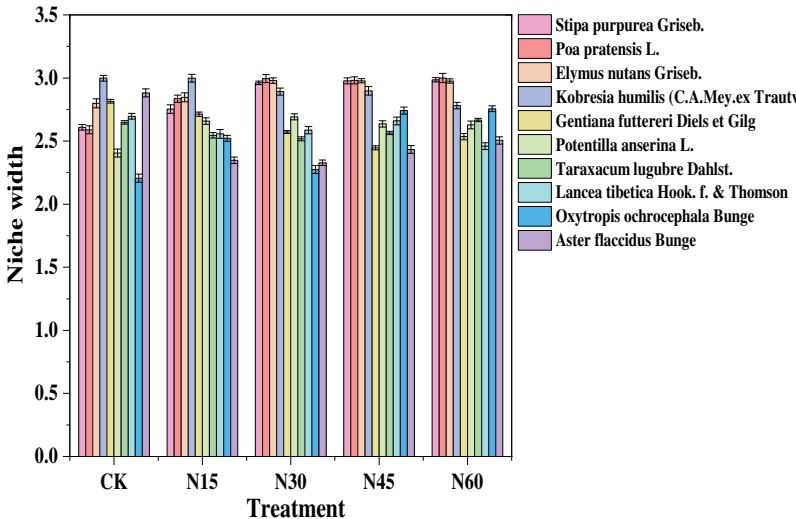

**Figure 2 Niche width of major plant species under different nitrogen addition levels.**

increased significantly by 11.36% under the N30 treatment but decreased significantly by 14.48% under the N60 treatment (F = 7.71, $p < 0.05$). Similarly, the species richness index increased significantly by 30.77% under the N30 treatment and decreased significantly by 23.08% under the N60 treatment (F = 11.72, $p < 0.05$). However, the Pielou evenness index did not show significant changes across different nitrogen levels (F = 0.81, $p > 0.05$) (Figs. 1A–1C).

Additionally, the importance values of plant functional groups varied with nitrogen addition levels. Compared to CK, the importance value of grasses increased significantly by 27.90%, 69.83%, and 153.96% under the N30, N45, and N60 treatments, respectively (F = 256.21, $p < 0.05$). In contrast, the importance value of sedges decreased significantly by 43.08% and 90.93% under the N45 and N60 treatments, respectively (F = 434.97, $p < 0.05$). The importance value of forbs, however, did not show significant changes under different nitrogen levels (F = 3.41, $p > 0.05$) (Fig. 1D).

Overall, these results indicate that with increasing nitrogen addition, both the Shannon-Weiner index and species richness index initially increased and then decreased, while the importance value of grasses showed a continuous increase and that of sedges a continuous decrease. The Shannon-Weiner index and species richness index reached their maximum values at a nitrogen addition level of 30 g N m$^{-2}$, while the importance value of grasses peaked at 60 g N m$^{-2}$.

## Niche characteristics

Niche width of the ten main species in the study area varied with different levels of nitrogen addition (Fig. 2). Without nitrogen addition, the species with the largest niche width were *Kobresia humilis* (3.00), *Aster flaccidus* (2.88), and *Elymus nutans* (2.80). At a nitrogen addition rate of 15 g N m$^{-2}$, the species with the largest niche width were *Kobresia humilis* (3.00), *Elymus nutans* (2.85), and *Poa pratensis* (2.84). With 30 g N m$^{-2}$ nitrogen

addition, the species with the largest niche width were *Poa pratensis* (3.00), *Elymus nutans* (2.98), *Stipa purpurea* (2.96), and *Kobresia humilis* (2.90). At 45 g N m$^{-2}$, the species with the largest niche width were *Poa pratensis* (2.98), *Elymus nutans* (2.98), *Stipa purpurea* (2.98), and *Kobresia humilis* (2.90). With 60 g N m$^{-2}$ nitrogen addition, the species with the largest niche width were *Poa pratensis* (3.00), *Stipa purpurea* (2.99), *Elymus nutans* (2.98), and *Kobresia humilis* (2.78). This indicates that at nitrogen addition rates of 30, 45, and 60 g N m$^{-2}$, the niche width of *Poa pratensis*, *Elymus nutans*, and *Stipa purpurea* exceeds that of sedges and other grasses.

Under different nitrogen addition levels, the niche overlap values for the ten major plant species exhibit variability. When the niche overlap value exceeds 0.83, there are 25 pairs (55.56%) without nitrogen addition, 19 pairs (42.22%) with N15, 21 pairs (46.67%) with N30, 17 pairs (37.78%) with N45, and four pairs (8.88%) with N60. For overlap values between 0.66 and 0.83, there are 10 pairs (22.22%) without nitrogen addition, nine pairs (20.00%) with N15, eight pairs (17.78%) with N30, 20 pairs (28.89%) with N45, and three pairs (6.67%) with N60. When the overlap value ranges from 0.50 to 0.66, there are five pairs (11.11%) without nitrogen addition, 11 pairs (24.44%) with N15, five pairs (11.11%) with N30, two pairs (4.44%) with N45, and four pairs (8.89%) with N60. For overlap values between 0.33 and 0.50, there are two pairs (4.44%) without nitrogen addition, three pairs (6.67%) with N15, six pairs (13.33%) with N30, two pairs (4.44%) with N45, and four pairs (8.89%) with N60. When the niche overlap value is between 0.16 and 0.33, there are three pairs (6.67%) with N15, five pairs (11.11%) with N30, five pairs (11.11%) with N45, and three pairs (6.67%) with N60. When the overlap value is less than 0.16, there are three pairs (6.67%) without nitrogen addition, six pairs (13.33%) with N45, and 27 pairs (60.00%) with N60 (Fig. 3). Overall, as the nitrogen addition increases, the number of plant species pairs with a niche overlap value below 0.16 progressively increases (Fig. 4).

## Characteristics of interspecific association
### Overall interspecific association
The overall inter-species connectivity (VR) varies with different nitrogen addition levels. When VR > 1, species in the plant community exhibit a positive correlation; when VR < 1, they exhibit a negative correlation; and when VR = 1, there is no correlation. As shown in Table 2, positive correlations are observed under CK, N15, and N30 treatments, while negative correlations are observed under N45 and N60 treatments. The W statistic (W = N × VR) is used to further test the significance of deviations of VR from 1. When the condition $X^2_{0.95}(N) < W < X^2_{0.95}(N)$ is met, the differences between species are not statistically significant; otherwise, significant associations exist. This indicates that under the N60 treatment, the overall inter-species connectivity is significantly negatively correlated, whereas under CK, N15, N30, and N45 treatments, there are no significant correlations among plant species (Table 2).

### Analysis of the interspecific relationship of main species
Figure 5 illustrates that, under different nitrogen levels (CK, N15, N30, N45, N60), the number of species pairs with an association coefficient (AC) ≥ 0.01 are 36, 32, 35, 27, and

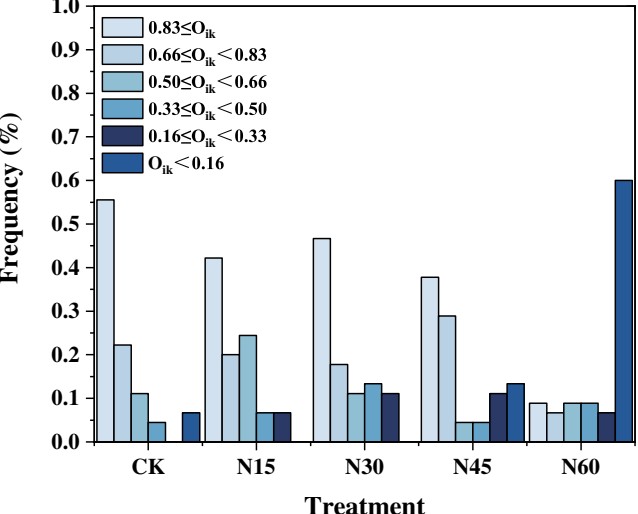

**Figure 3 Niche overlap values of major plant species under different nitrogen addition levels.**

**Figure 4 Distribution frequencies of niche overlap values under different nitrogen addition levels.**

**Table 2 The impact of different nitrogen levels on inter-species associations in plant communities.**

| Treatment | N | S | Var. Ratio | Statistics W | Chi-square critical value ($\chi^2_{0.95}$ (N) ≤ $\chi^2$ ≤ $\chi^2_{0.05}$ (N)) | Community relevance |
|---|---|---|---|---|---|---|
| CK | 3 | 10 | 1.20 | 3.60 | (0.35, 7.82) | Positive correlation |
| N15 | 3 | 10 | 1.33 | 3.99 | (0.35, 7.82) | Positive correlation |
| N30 | 3 | 10 | 1.47 | 4.41 | (0.35, 7.82) | Positive correlation |
| N45 | 3 | 10 | 0.82 | 2.46 | (0.35, 7.82) | Negative correlation |
| N60 | 3 | 10 | 0.11 | 0.33 | (0.35, 7.82) | Significant negative correlation |

**Note:**
In the table, CK, N15, N30, N45, and N60 correspond to nitrogen additions of 0, 15, 30, 45, and 60 g N·m$^{-2}$, respectively. $N$ and $S$ represent the total number of plots and species, respectively.

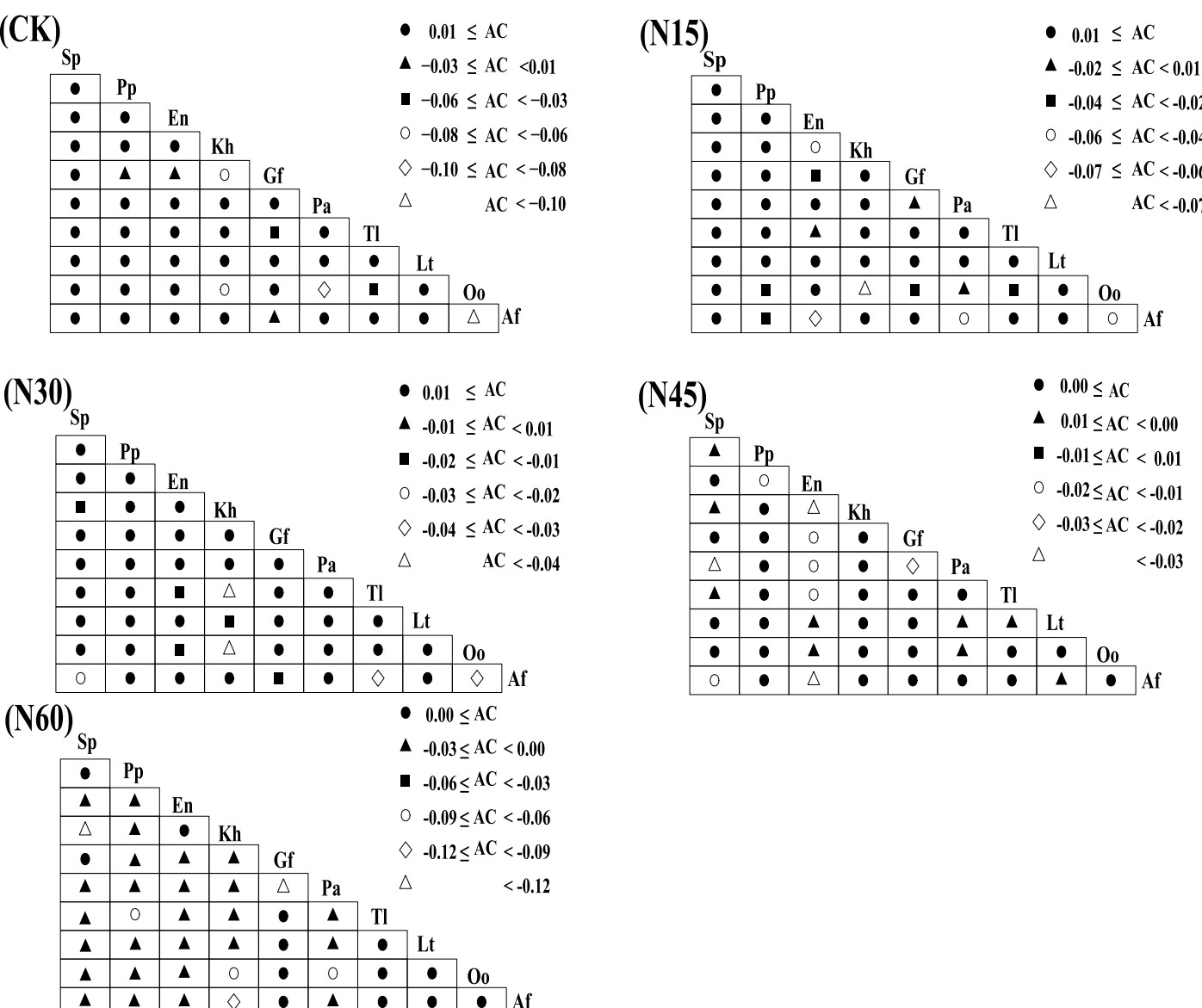

**Figure 5 The impact of different nitrogen levels on interspecies connectivity within plant communities.**

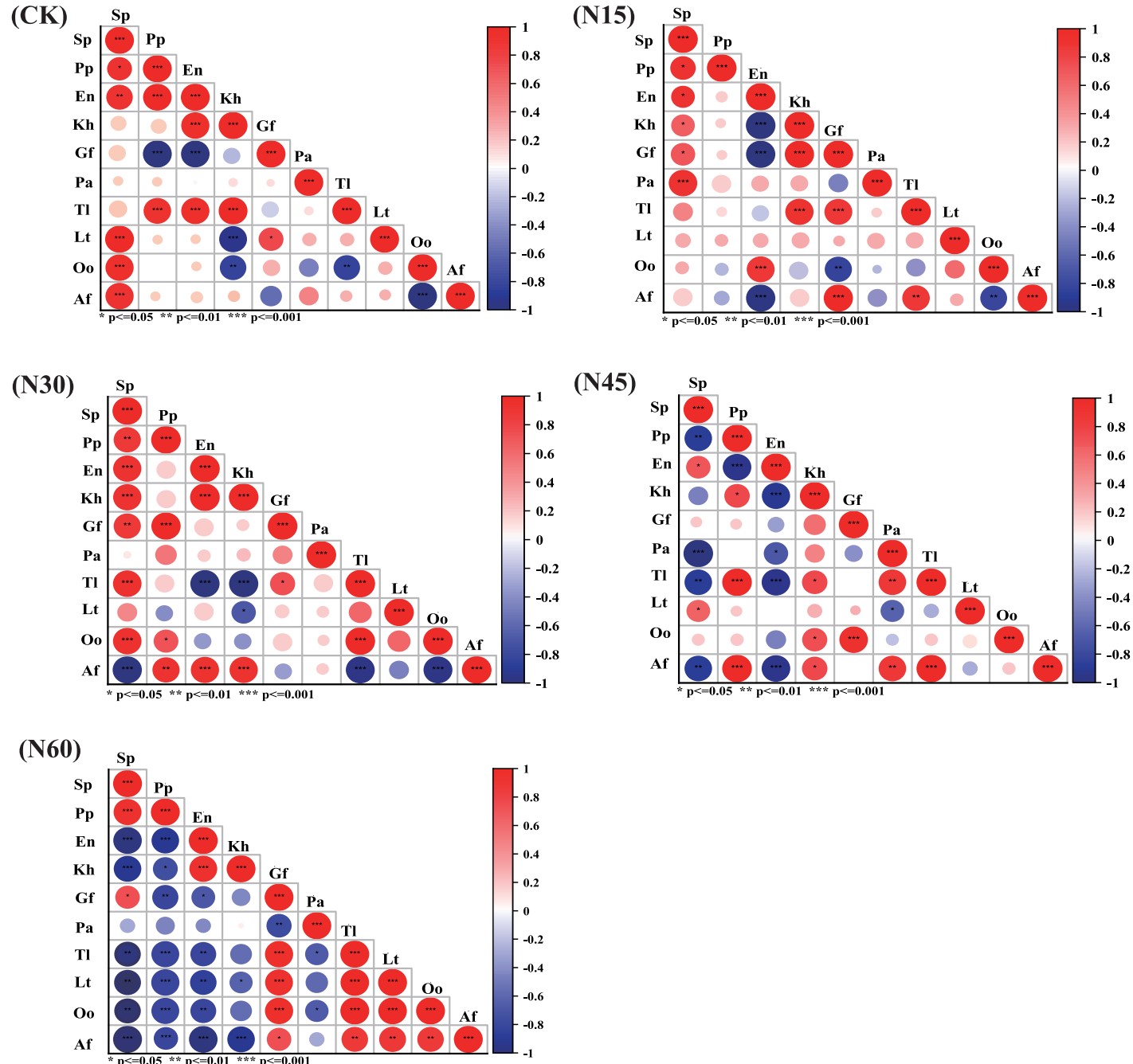

**Figure 6 The effect of different nitrogen levels on Spearman correlation coefficients and their significance.** The color depth of the circle in the figure represents the size of the correlation coefficient. The lighter the color, the less linear correlation between the two variables. The size of the circle represents the absolute value of the correlation coefficient. The larger the area, the greater the absolute value. * Indicates $p < 0.05$; ** indicates $p < 0.01$; *** indicates $p < 0.001$.

22, respectively, while those with AC < 0.01 are 9, 13, 10, 18, and 23, respectively. The plant interspecies relationships significantly vary with nitrogen addition levels (Fig. 6). Under the CK, N15, N30, N45, and N60 treatments, the number of significantly positively correlated species pairs are 11, 11, 14, 12, and 13, respectively, while the number of

significantly negatively correlated pairs are 6, 5, 6, 10, and 23, respectively. Specifically, under the CK treatment, *Stipa purpurea* shows significant positive correlations with *Lancea tibetica*, *Oxytropis ochrocephala*, and *Aster flaccidus*; *Poa pratensis* is significantly positively correlated with *Taraxacum lugubre*; and *Elymus nutans* is significantly positively correlated with *Kobresia humilis* and *Taraxacum lugubre*. Under the N15 treatment, *Stipa purpurea* shows significant positive correlations with *Kobresia humilis*, *Gentiana futtereri*, and *Potentilla anserina*; and *Elymus nutans* is significantly positively correlated with *Oxytropis ochrocephala*. Under the N30 treatment, *Stipa purpurea* shows significant positive correlations with *Kobresia humilis*, *Gentiana futtereri*, *Taraxacum lugubre*, and *Oxytropis ochrocephala*; *Poa pratensis* is significantly positively correlated with *Gentiana futtereri*, *Oxytropis ochrocephala*, and *Aster flaccidus*; and *Elymus nutans* is significantly positively correlated with *Kobresia humilis* and *Aster flaccidus*. Under the N45 treatment, *Stipa purpurea* shows significant negative correlations with *Potentilla anserina*, *Taraxacum lugubre*, and *Aster flaccidus*, but a significant positive correlation with *Lancea tibetica*; *Poa pratensis* is significantly positively correlated with *Kobresia humilis*, *Taraxacum lugubre*, and *Aster flaccidus*; while *Elymus nutans* shows significant negative correlations with *Kobresia humilis*, *Potentilla anserina*, *Taraxacum lugubre*, and *Aster flaccidus*. Under the N60 treatment, *Stipa purpurea* is significantly positively correlated with *Gentiana futtereri* and negatively correlated with *Kobresia humilis*, *Taraxacum lugubre*, *Lancea tibetica*, *Oxytropis ochrocephala*, and *Aster flaccidus*; *Poa pratensis* shows significant negative correlations with *Kobresia humilis*, *Gentiana futtereri*, *Taraxacum lugubre*, *Lancea tibetica*, *Oxytropis ochrocephala*, and *Aster flaccidus*; while *Elymus nutans* is significantly positively correlated with *Kobresia humilis* and negatively correlated with *Gentiana futtereri*, *Taraxacum lugubre*, *Lancea tibetica*, *Oxytropis ochrocephala*, and *Aster flaccidus*. Overall, with increasing nitrogen levels, grasses (*Stipa purpurea*, *Poa pratensis*, and *Elymus nutans*) show significant negative interspecies correlations with other sedges and forbs.

# DISCUSSION

## Plant species diversity

The plant species diversity index is an effective metric for assessing plant community heterogeneity and successional processes (*Wei et al., 2024*). The results of this study indicate that, with increasing nitrogen addition levels, both the Shannon-Weiner index and the species richness index initially increase and then decrease. This trend is consistent with findings from temperate grassland studies, which show that both low and high-nitrogen treatments significantly reduce plant species richness (*Zhang et al., 2014*). The addition of an optimal amount of nitrogen can lead to differences among plant species in nitrogen utilization, growth rate, and life cycle, resulting in greater ecological niche differentiation and enabling more species to coexist in the same habitat (*Tian et al., 2022*). However, excessive nitrogen addition may enhance the nitrogen utilization abilities of certain plants, leading to increased interspecific competition. This competition is a key factor in the reduction of plant diversity under high nitrogen conditions, as dominant species compete for available nitrogen, suppressing the growth of smaller species and

reducing niche dimensions (*Farrer & Suding, 2016*). Our findings suggest that at a nitrogen application rate of 30 g N m$^{-2}$, both the Shannon-Weiner index and species richness index reach their peak values. This is attributed to the fact that nitrogen addition at a moderate level (272.24 kg ha$^{-1}$ yr$^{-1}$) eliminates nitrogen limitations in degraded grasslands, stimulates community growth, and thereby enhances community stability (*He et al., 2024*).

Many scholars consider increased nitrogen deposition to be a significant factor influencing plant community composition and diversity (*Han et al., 2019*; *Zhou et al., 2018*). Due to varying nitrogen utilization strategies and efficiencies, different species and functional groups respond differently to nitrogen deposition, leading to shifts in plant community composition (*Kwaku et al., 2021*). Research in temperate grasslands has shown that long-term nitrogen enrichment reduces the dominance of graminoids and herbaceous plants while increasing the dominance of annual plants (*Bai et al., 2010*). Some studies suggest that nitrogen deposition allows dominant species to acquire more nutrients for rapid growth, thereby decreasing plant community diversity by reducing the abundance of rare species (*Zhang et al., 2022b*). The results of this study indicate that as nitrogen addition increases, the importance value of grasses gradually rises, while the importance value of sedges progressively declines. This is primarily because 1) intense competition within plant communities drives nitrogen-preferring plants to access lighter and water resources. Compared to sedges, grasses such as *Stipa purpurea*, *Poa pratensis*, and *Elymus nutans* are nitrogen-loving species (*Zhang et al., 2015*). Furthermore, dominant functional groups or species capture soil resources more rapidly after nitrogen enrichment, leading to increased importance values for grasses (*Liu et al., 2018*). 2) Nitrogen addition enhances plant productivity and canopy height, shifting competition from below-ground nutrient acquisition to above-ground light competition (*DeMalach, Zaady & Kadmon, 2017*). Faster-growing or taller species capture more light than shorter or slower-growing species, intensifying competitive exclusion among species and reducing the importance values of sedges. In summary, different plant functional groups respond differently to nitrogen, leading to changes in plant community composition and structure.

## Niche characteristics

Niche width reflects a species' adaptability to environmental conditions and its capacity to utilize ecological resources (*Fajardo & Siefert, 2019*). A larger niche width indicates a greater ability to exploit resources (*Yang et al., 2021*). This study reveals that, at nitrogen addition levels of 30, 45, and 60 g N·m$^{-2}$, the niche widths of *Poa pratensis*, *Elymus nutans*, and *Stipa purpurea* are greater than those of other plant taxa. This is because tall graminoids in grasslands possess deeper root systems and exhibit stronger competitive advantages for light and nutrients, which enhances their dominant position in plant communities (*Zhao et al., 2023*). The study also found that species with the largest niche widths are not confined to graminoids; certain forbs (*Aster flaccidus*, *Oxytropis ochrocephala*, *Gentiana futtereri*) and sedges (*Kobresia humilis*) also have considerable niche widths. This indicates that these species have strong ecological adaptability and hold a significant competitive advantage in spatial resource utilization.

In conditions of resource scarcity, niche overlap reflects both the ecological similarity between species and the level of competition among them. Conversely, under resource-rich conditions, niche overlap merely indicates ecological similarity or the occupation of similar ecological space by species (*Wang et al., 2023*). The results of this study indicate that as nitrogen addition increases, the niche overlap values also show a rising trend. Previous research also shows that nitrogen addition significantly reduces niche overlap among plant species (*Du et al., 2024*). This can be attributed to: 1) an increase in nitrogen addition levels leading to dominance by nitrogen-preferring species such as *Poa pratensis* and *Elymus nutans*. These species share similar ecological characteristics and resource utilization strategies, resulting in a more concentrated niche space and consequently reduced niche overlap among species (*Yang & Hui, 2021*). 2) Nitrogen addition induces significant differences in resource utilization among some plants, leading to niche differentiation. This reduces resource competition and promotes species co-occurrence, thereby decreasing interspecies niche overlap (*Tsafack et al., 2021*).

Research has found that species with larger niche widths tend to exhibit greater niche overlap with other species (*Yuan et al., 2016*). In this study, *Elymus nutans* demonstrated a large niche width and also showed considerable niche overlap with *Oxytropis ochrocephala* and *Aster flaccidus*. This is because species with larger niche widths are better able to utilize environmental resources and have broader distribution ranges, resulting in higher probabilities of niche overlap with other species (*Liu et al., 2020*). Conversely, species such as *Potentilla anserina* and *Oxytropis ochrocephala* have smaller niche widths but show high niche overlap values. This may be due to differences in resource utilization strategies among plants, leading to some degree of niche differentiation (*Pulla et al., 2017*).

## Characteristics of interspecific association

The associations between species are primarily influenced by positive correlations arising from mutualistic or symbiotic relationships, and negative correlations resulting from competition for shared resources (*Wang et al., 2023*). Positive correlations indicate that species have similar or identical environmental resource requirements, leading to strong complementarity and more efficient resource utilization (*Jin et al., 2022*). Conversely, negative correlations suggest significant differences in biological traits and adaptations to environmental heterogeneity, resulting in niche separation (*Pastore et al., 2021*). The results of this study reveal that in the CK, N15, N30, and N45 treatments, most species pairs showed non-significant correlations. Previous research indicates that when species have relatively independent distribution patterns within a community, and the likelihood of encountering other species is low, most species pairs exhibit weak or no associations (*Yuan & Wang, 2023*). This phenomenon is attributed to reduced species richness and lower individual density of herbaceous plants following nitrogen addition, leading to increasingly independent distributions and a non-associative trend among populations (*Zhang et al., 2022a*). In contrast, under the N60 treatment, the overall connectivity between plant species showed significant negative correlations. Previous studies also suggest that nitrogen addition disrupts the original stability of plant communities, leading to intensified competition for nutrients among species and a shift towards negative overall connectivity

(*Du et al., 2024*). Additionally, increased levels of soil nutrients (such as moisture, organic carbon, and nitrogen) exacerbate competition for survival resources among species, including competition for above-ground light resources, resulting in significant negative interspecies correlations (*Juan et al., 2023*). Research has also shown that nitrogen addition disrupts plant species composition and reduces plant diversity, primarily due to competitive interactions among dominant species (*Du et al., 2024*). In this study, at a nitrogen addition level of 60 g N·m$^{-2}$, grasses (*e.g.*, *Stipa purpurea*, *Poa pratensis*, and *Elymus nutans*) exhibited significant negative correlations with sedges (*Kobresia humilis*) and forbs (*Gentiana futtereri*, *Taraxacum lugubre*, *Lancea tibetica*, *Oxytropis ochrocephala*, and *Aster flaccidus*). This indicates that nitrogen addition intensifies competition between dominant grasses and other species, disrupting existing balances and reducing plant diversity, leading to a less stable plant community.

## CONCLUSIONS

Under nitrogen addition, grasses exhibit significantly wider niches than other plant groups, and niche overlap values decrease, with overall interspecies connectivity showing a negative correlation. This suggests that high levels of nitrogen addition in the alpine meadows of the Tibetan Plateau exacerbate competition between grasses, sedges, and other forbs. The strong ecological adaptability of grasses leads to an expanded niche width and reduced niche overlap, disrupting the existing balance and diminishing plant community diversity. These findings provide crucial data and theoretical insights for understanding the competitive mechanisms and changes in plant community diversity under high nitrogen levels in alpine meadows. Furthermore, exploring the role of other environmental factors, such as soil moisture and temperature, in shaping plant community responses to nitrogen deposition will offer a more comprehensive understanding of future shifts in community structure and composition under climate change. This research will contribute to the development of biodiversity conservation strategies and sustainable management practices for alpine meadows on the Tibetan Plateau.

### Funding

This study was supported by the National Key Research and Development Project (2022YFD1602302), the Key R&D and Transformation Plan of Qinghai Provincial Science and Technology Department (2024-NK-137), and the Sanjiangyuan Ecosystem Field Observation and Research Station of the Ministry of Education (K9922050). The Research and demonstration on seed propagation and silage processing and storage technology of annual forage grass in Chengduo County (2024-NK-P28) supported the APC of this article. The funders had no role in study design, data collection and analysis, decision to publish, or preparation of the manuscript.

## Grant Disclosures

The following grant information was disclosed by the authors:

National Key Research and Development Project: 2022YFD1602302.

Key R&D and Transformation Plan of Qinghai Provincial Science and Technology Department: 2024-NK-137.

Sanjiangyuan Ecosystem Field Observation and Research Station of the Ministry of Education: K9922050.

The research and demonstration on seed propagation and silage processing and storage technology of annual forage grass in Chengduo County: 2024-NK-P28.

## Competing Interests

The authors declare that they have no competing interests.

## Author Contributions

- Xuemei Xiang conceived and designed the experiments, performed the experiments, analyzed the data, prepared figures and/or tables, authored or reviewed drafts of the article, and approved the final draft.
- Ke Jia De conceived and designed the experiments, prepared figures and/or tables, authored or reviewed drafts of the article, and approved the final draft.
- Weishan Lin performed the experiments, authored or reviewed drafts of the article, and approved the final draft.
- Tingxu Feng performed the experiments, authored or reviewed drafts of the article, and approved the final draft.
- Fei Li performed the experiments, authored or reviewed drafts of the article, and approved the final draft.
- Xijie Wei performed the experiments, authored or reviewed drafts of the article, and approved the final draft.

## Data Availability

The raw measurements are available in the Supplemental File.

## Supplemental Information

Supplemental information for this article can be found online at http://dx.doi.org/10.7717/peerj.18526#supplemental-information.

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
