# Peer review of "The ecological niche characteristics and interspecific associations of plant species in the alpine meadow of the Tibetan Plateau affected plant species diversity under nitrogen addition"

_PeerJ, doi:10.7717/peerj.18526_

## Round 0.1 · original submission · Major Revisions

Dear authors,

Two detailed and concise expert reviews suggest that your MS should be substantially revised for publication in PeerJ, and I fully agree with them. As your MS has a high potential to contribute to the relevant field of research, I'm sure that the improvements suggested by the Reviewers will greatly contribute to a better presentation of your work, so please consider and address all their comments carefully. In addition to all other issues, please pay great attention to the quality of your figures - they need significant improvement (resolution, size and explanations both in the figure and in the caption, which must be self-explanatory text) according to the specific comments of the reviewers.

Reviewer 1 ·

Basic reporting

The significance of plant species diversity in maintaining the structure and function of grassland ecosystems cannot be overstated. Analyzing community niches and interspecific associations is essential for comprehending fluctuations in plant species diversity. This study investigates the impact of different nitrogen addition levels on plant species diversity, functional group importance value, niche breadth, niche overlap, and interspecific associations in an alpine meadow on the Qinghai-Tibet Plateau. These findings provide valuable insights into the restoration of degraded alpine grasslands from the perspective of alpine meadow plant communities under nitrogen addition. Overall, the manuscript is well-structured, supported by ample data to substantiate the main conclusions, and effectively establishes a logical framework. However, major revisions should be presented before publication.

Experimental design

The experimental design is reasonable.

Validity of the findings

The manuscript is well-structured, supported by ample data to substantiate the main conclusions, and effectively establishes a logical framework.

Additional comments

1. Please Rephrase this sentence. Line 24.
2. The sentence "the importance value in which sense" appears to be incomplete or unclear. Please provide additional context or clarify it. Lines 25 and 26
3. Please add quantitative data to the results section of your abstract.
4. Please also include the implications of your study in the final 1-2 sentences of the abstract.
5. Please ensure that all terms are clearly defined. For instance, the term “Niche breath” might not be familiar to all readers. A brief definition or explanation of this term in the text would be helpful.
6. Incorporate the most recent references into the introduction section.
7. Please rephrase this sentence Line 105-106.
8. Why did you select higher nitrogen addition levels: 15 g N•m-2 (N15), 30 g N•m-2 (N30), 45 g N•m-2 (N45), and 60 g N•m-2? Please provide reason for use high N addition levels.
9. Every result should be discussed in the Discussion section, please confirm it.
10. Include additional findings in the conclusion section.
11. Please enhance the visibility and clarity of all figures for better understanding.
12. Incorporate future perspectives on the study in the conclusion part.

Annotated reviews are not available for download in order to protect the identity of reviewers who chose to remain anonymous.

Reviewer 2 ·

Basic reporting

• Clear and unambiguous, professional English used throughout.
o Overall, the authors use clear and professional English throughout the manuscript, though there were some instances where this could be improved. For example, line 24 requires clarity around a single index vs all three diversity indices and the use of ‘nevertheless’ on line 44 is not appropriate in this context. Line 60 also requires grammatical corrections. I suggest having a colleague who is proficient in English to read over the manuscript to correct small errors such as these.
o Some formatting inconsistencies around scientific notation need addressing on lines 22-24 and line 156.

• Literature references, sufficient field background/context provided.
o The manuscript would benefit greatly from a significant increase in the amount of background information provided. In the introduction, providing more detailed explanations of what the niche is and what you mean by the terms niche breadth, niche overlap and interspecific association, plus terms relating to these would be beneficial. I would also urge the authors to be consistent in their usage of these terms, rather than switching between variations, e.g., niche breadth vs niche width and inter-species vs interspecific competition. Please see the specific examples below and suggestions for correction.
Lines 65-66: Niche breadth and niche overlap need explaining. There are many different ways in which the “niche” has been defined, and it would be good to clarify which you refer to. Equally important, is to define what you mean by “breadth” and “overlap”.
Lines 69-70: This is a good point to make, though again, clarifying what “niche dimensionality” means is important here. Perhaps these studies should also be related to the concepts of breadth and overlap you have just mentioned.
Lines 73-34: I think it would be beneficial for clarity to consolidate your wording of “inter species competition”, “interspecific competition”, “species competition” to a single term, especially since you haven’t defined what this means in the introduction (this would also be beneficial).
Line 75: You define interspecific association, although not in enough detail. You should specifically define the difference between interspecific association and interspecific interactions.
Line 81: Ecosystem degradation should be defined as it is an ambiguous term.
Line 85: What you mean by “direction and intensity” needs to be defined.
Lines 89-92: It is unclear to suggest that ‘maintaining biodiversity’ is an ecosystem service without further clarification.
o There are a few instances where the appropriate context for the statement given is not provided, or they are very loosely associated. For example, on line 58 the citation of Grime 1973 is inappropriate as the text refers to studies which demonstrate that declining richness under high nitrogen is caused by increased biomass and therefore, heightened interspecific competition. The references should therefore be more directly related to the text.

• Professional article structure, figures, tables. Raw data shared.
o The structure of the manuscript conforms well to the requirements of this journal and the raw data is shared.
o The figures require significant improvement before publication. For all figures, the resolution is much too low to be easily readable and many require more detail in the captions to allow the reader to understand them. Please see specific suggestions below:
Figure 1: The figure caption should include descriptions of what each facet a-d represents. There is also no indication of what the error bars refer to, e.g., standard error, 95% CI. In 1d, there are three new groupings used. It is unclear as to what these groups are, given the ambiguous naming of the group 'Grasses' earlier in the text.
Figure 2: It is unclear what 'main' species are. What level of coverage makes it a 'main' plant? This should be specified in the text.
Figure 3: Given the low resolution this figure is un-interpretable in respect to the symbol correlations with niche overlap values. Consider including the species labels on the figure, with names at a 45 degree angle, or name abbreviations. Alternatively, a legend in the blank panel containing the species number-to-name IDs. This is easier to interpret than listing them in the figure caption.
Figure 4: There needs to be some explanation of what this figure is showing as it's very unclear. You should label each individual figure a-j and clearly explain what the left and right-hand panels are showing.
Figure 5: What the circle size represents is unclear. The right-hand side of each panel should be labelled as 'Spearman Correlation Coefficient' for clarity.
Table 1: Further clarification as to the results presented here is needed in the text. Re-iterating the meaning of some terms in the table caption would also be helpful.
o The raw data is provided, although it does not contain a metadata sheet, explaining clearly how each term was calculated and what each column or row represents. There are two importance value columns containing different numbers and these require explanation.

• Self-contained with relevant results to hypotheses.
o All results relevant to the hypotheses are included.

Experimental design

• Original primary research within Aims and Scope of the journal.
o The authors are commended for providing original research that meets the journal requirements.

• Research question well defined, relevant and meaningful. The knowledge gap being investigated should be identified, and statements should be made as to how the study contributes to filling that gap.
o The two research hypotheses provided on lines 98 – 100 are relevant and meaningful but are not well-defined. Specifically, for question 2, it is unclear how you will be linking variation in niche characteristics and interspecific associations back to diversity metrics. Additional background information as suggested above should address this issue. For example, on line 77 you mention the integration of niche and interspecific association research, but don’t provide sufficient information about how these two areas relate to one another in previous research or your own, nor what we will specifically gain by combining the two fields. This is made difficult due to the lack of clarity around the indices of niche breadth and overlap, and interspecific association.

• Rigorous investigation performed to a high technical and ethical standard.
o Some technical aspects require clarification to be compared to previous work in this field. Specifically, the grouping of plants for analyses should be reconsidered.
On lines 136-138, you detail the functional groupings, though they are not consistent with standard groups used in the field. Given that the Poaceae, or Gramineae, group are commonly referred to as grasses, having the third group named as ‘grasses’ will be confusing for readers. As there is no supplementary information showing how your species are grouped (this would be a good consideration) it is unclear whether it means that all species in this meadow are 'graminoids' i.e., grasses, rushes or sedges, or whether the 'grasses' grouping refers to all other herbaceous plants at the site, including forbs. A clearer system, and one used commonly in the field (see Li et al. 2021 Frontiers in Plant Science) uses grasses (Poaceae species), sedges (Cyperaceae species) and forbs (usually herbaceous dicots). Often, ‘rushes’ (Juncaceae species) and ‘woody plants’ (non-herbaceous, woody species) are also included, depending on the vegetation type.
o In figure 1d, you use the groupings (forbs, sedge, grasses) with no clarification as to what these groups are given in the figure caption nor elsewhere in the text. For consistency, it would be best to define these groups in the methods and consistently use it throughout the manuscript.
o The term ‘main plants’ is used in hypothesis 2 on line 100, on line 260 and in figure 2, but it is never defined in the manuscript. This should specifically relate to something like species biomass or abundance and must be specified.

• Methods descried with sufficient detail and information to replicate.
o When describing the sampling method for pants species you use the ‘square grid’ method on line 130, though this needs to be defined. You should specify the number of grid cells for clarity of sampling effort. Also, specify whether you did a point-cover estimate vs a presence/absence per grid cell estimate vs visual estimation. If the ‘square grid’ method covers these issues, please add a reference which explains the method in greater detail.
o Information about the site conditions could use further clarity. For example, if you provide a range of annual mean temperatures as on line 106, you should also provide an indication of the timeframe over which these measurements occurred. E.g., the annual average temperature on the plateau has ranged from -5.6 to 3.8 degrees C over the last 50 years. Otherwise, it's unclear how the annual mean is varying (it could be annual mean minimum and maximum temperature).

Validity of the findings

• Impact and novelty not assessed. Meaningful replication encouraged where rationale and benefit to literature is clearly stated.
o The findings of this study in terms of the benefit to the literature are provided in the introduction.
• All underlying data have ben provided; they are robust, statistically sound, and controlled.
o The statistical basis of the results presented in this manuscript require further clarification by the author for full interpretability. This will be helped by increased clarity of resolution in figure 5 and a more detailed caption as discussed above. Also, the result on line 214 does not report the p value for each of the ANOVAs, rather only that they were all <0.05, so true significance values for each are not provided.
• Conclusions are well stated, linked to original research question and limited to supporting results.
o The authors have made the answer to question 1 quite clear, though question 2 is still unclear. The discussion does not sufficiently link the results together for the reader, thus the relevance of each individual result is not obvious. It is clear that diversity has been altered, but exactly how is obscured by ambiguous functional groupings. Similarly, the link between the niche characteristics investigated, interspecific association and diversity is not made clear to the reader.

Additional comments

I commend the authors for the effort put into preparation of this manuscript. It has a lot of potential as the underlying research questions are interesting and the experimental design is sound, however at this stage, many aspects remain ambiguous. My detailed comments are above, though I hope that an additional, more general summary here is also helpful.
The most important issue to address in this manuscript is the inclusion of more background information about the niche and interspecific associations and their definitions. Specifically, you must address the benefits of considering both of these topics together and clarify their relevance to diversity - this will help you to define your second research question. The second most important issue would be to provide context as to the types of analyses selected, how they have been used to effectively answer questions like this in the past and why they are the best method to use here, e.g., your importance values and association coefficients. The next most important issue is clarity around the functional groupings of your species, without which the reader is unable to determine the meaning of your results. Lastly, your discussion should synthesise the different results to clearly explain to the reader how they relate to one another and how they should be interpreted, with appropriate references to the literature.
Once these issues have been addressed this manuscript has great potential for publication.

---

## Round 0.2 · Minor Revisions

Thank you for this significantly improved R1 version of your MS.
I acknowledge the changes to the figures to make them clearer and more reader friendly.
I have asked the Reviewer of your R0 version to review your R1 and I fully agree with the new remarks and suggestions.
In addition to the Reviewer's points, I'd like to request that you correct as follows:
1) Niche breadth vs. Niche width: please be consistent in your use of these terms, you still seem to use them interchangeably. For example, you give the same definition but use the two terms in lines 77-78 for breadth and 369-370 for width (line numbers refer to the resubmitted document). Also, in Figure 2, the caption uses the term niche breadth and the Y-axis title uses niche width. Please correct throughout the MS.

2) Figure 1. The caption is a self-explanatory text, so you should add here what is presented in (a) layer, (b) layer etc.. The caption should also state that the different letters indicate statistically significant differences between N treatments at P < 0.05.

Reviewer 2 ·

Basic reporting

*Please note that line numbers refer to the re-submitted document, not the track-changed version
Nicely written with few grammatical errors. The introduction shows significant improvement and is much clearer. The figure quality is now great and are mostly well-labelled and described. There are still a few things that I believe should be addressed before pubication, most importantly is the re-wording of some results. The re-write of the plant species diversity component is very good, and I appreciate the summary sentences at the end. The ‘characteristics’ section of results should be looked at similarly. It would be clearer if you were to highlight the most important results only in the text and use a table or figure for the specifics. For example, the written sentence on lines 267-268 and lines 282-284 provide a nice summary of these results; highlighting more summary information like this and deferring the rest to a table or figure would improve readability and reader engagement. This applies to the ‘Analysis of the interspecific relationship of main species’ section of results also. Great work improving the clarity of the ‘Overall interspecific association section’.
Further small improvements could be made for the below:
Line 29: Please remove ‘they’.
Line 87-89: This sentence is still not capturing what interspecific association is and niche characteristics are not well-defined. I think it would be better to remove this sentence of description and use something simpler, like: “niche characteristics (breadth and overlap)”, to make it clearer. You should specify the link between an interspecific association and an interaction, because one implies the other but does not directly measure it and you should be upfront about the approach you used.
Line 96-100: If you are going to make a definitive statement about interactions and stability like this, you should have more than one reference from a single system. If supporting data from multiple systems doesn’t exist, I would change the wording.
Line 357-358: Point 1) needs re-wording.
Lines 406-412: Please reconsider these lines as they are contradictory to one another. Species associations are driven by both the processes you mention, but they are stated in such a way as to seem mutually exclusive.
Figure 6: This still requires further labelling of the coloured bar for clarity and there is no indication as to what circle size represents.

Experimental design

The research questions are now well-defined and I have a better understanding of the rationale behind your project and where it fits into the knowledge gap.
The re-classification of your groupings is clear and much more interpretable. I would only advise that ‘weeds’ be changed to ‘forbs’ as this is much more common and ‘weeds’ often indicate non-nativeness, so may give the incorrect idea to readers.
Throughout the manuscript you refer to the niche overlap of species with reference to the value 0.16, but I can’t find where you explain the significance of this particular number anywhere. It would be useful to clarify this. What’s more, in the discussion, you should move away from these specific result-reporting numbers and report the significance of what happened instead (i.e., niche overlap was reduced or increased).

Validity of the findings

The conclusions are excellent, although I don’t think a summary of the results (the first three sentences) is necessary here, it should be integrated with the significance and meaning as you do in the sentences following.

---

## Round 0.3 · Minor Revisions

Your R2 Word file-with changes tracked is not identical with the R2 pdf file produced (for some reason). Please correct the following in your Word file and check the final pdf produced, before submitting.

There are still some “niche breadth” in your text (Lines 85, 86) and captions (Figure 2). Also, in the old version of the Abstract that appears first in you R2 pdf. Please use the “find-replace” tool of Word to track and correct them and refresh your Abstract in the Peerj submission system.

Finally, please check and correct the following grammatical errors:
1) the spaces between words in the revised caption of Figure 1.
2) Line 257: Paragraph at “The…”
3) Line 369: Capitalize the first letter of the paragraph
4) Line 376: “forbs” is doubled.

---

## Round 0.4 · accepted · Accept

You have not replaced "breadth" with "width" in the caption of Figure 2 as I asked you. You will be have the opportunity to do so in the final production stage.